

**Correcting for trace gas absorption when retrieving aerosol optical**
**depth from satellite observations of reflected shortwave radiation**
Falguni Patadia[1,2], Robert C. Levy[2], Shana Mattoo[2,3]
[1]GESTAR/Morgan State University, Columbia, MD, USA
[2]NASA Goddard Space Flight Center, Greenbelt, MD, USA
[3]SSAI, Lanham, MD, USA



## Abstract

Retrieving aerosol optical depth (AOD) from top-of-atmosphere (TOA) satellite-measured radiance requires separating the aerosol signal from the total observed signal. Total TOA radiance includes signal from underlying surface and from atmospheric constituents such as aerosols, clouds and gases. Multispectral retrieval algorithms, such as the dark-target (DT) algorithm that operates upon Moderate Resolution Imaging Spectroradiometer (MODIS, onboard Terra and Aqua satellites) and Visible Infrared Imaging Radiometer Suite (VIIRS, onboard Suomi-NPP) sensors, use wavelength bands in "window" regions. However, while small, the gas absorptions in these bands are non-negligible and require correction. In this paper we use High-resolution TRANsmission (HITRAN) database and Line-by-Line Radiative Transfer Model (LBLRTM) to derive consistent gas corrections for both MODIS and VIIRS wavelength bands. Absorptions from $H_2O$, $CO_2$ and $O_3$ are considered, as well as other trace gases. Even though MODIS and VIIRS bands are "similar", they are different enough that applying MODIS specific gas corrections to VIIRS observations results in an underestimate of global mean AOD (by 0.01), but with much larger regional AOD biases up to 0.07. As recent studies are attempting to create a long-term data record by joining multiple satellite datasets, including MODIS and VIIRS, the consistency of gas correction becomes even more crucial.



## 1. Introduction

Aerosols are tiny particles in the atmosphere that scatter and/or absorb incoming solar
insolation, and because of this are active players in Earth's energy budget [*IPCC*, 2013]. In
addition aerosols affect cloud and precipitation processes [*Denman et al.*, 2007; *Boucher et al.*,
2013], and they degrade air quality, contributing to increased morbidity and mortality rates
world-wide [*Lim et al.* 2012]. For these reasons characterizing and monitoring aerosol
distributions has become a global priority [*Boucher et al.*, 2013].
Satellite aerosol remote sensing allows for characterization and monitoring of aerosols
over broad regions and globally [*Lenoble et al.,* 2013]. Different aerosol remote sensing
schemes are applied, depending on the information received by the different satellite sensors
[*McCormick et al.*, 1979; *Herman et al.*, 1997; *Stowe et al.*, 1997;*Tanré et al.*, 1997; *Kaufman et*
*al.*, 1997a; *Torres et al.*, 1998; *Veefkind et al.*, 1998; *Higurashi and Nakajima*, 1999; *Deuzé et*
*al.,* 1999; *Knapp et al.*, 2002; *Martonchik et al.*, 1998; *Liu et al.*, 2005; *Kahn et al.*, 2001]. In
terms of passive satellite sensors that measure solar radiation reflected by the Earth-atmosphere
system, aerosol remote sensing methods must isolate the information obtained from the solar
radiation interacting with suspended aerosol particles from the information obtained from all
other interactions: reflectance from the surface, scattering from atmospheric molecules and
clouds, absorption by atmospheric gases, etc. [*Vermote et al*, 1997]. Thus, characterizing and
removing these other sources of information in the satellite signal becomes a fundamental part of
the process.
Some of the interactions requiring removal continue to receive considerable attention as
new sensors are deployed and new aerosol remote sensing algorithms are derived. These include
characterizing the contribution from the surface and masking clouds [*Hutchison et al.*, 2008; *Shi*





*et al*., 2014;]. Other interactions received much less attention, as these are considered to be well-
understood and simple to apply in new situations. These latter include molecular scattering and
gaseous absorption [*Tanré et al*., 1992; *Vermote et al*., 1997]. However, the requirements on
accuracy of aerosol remote sensing products become tighter as instrument capabilities,
calibration and retrieval methods improve. For example, *Hollman et al*., (2013) has recently
suggested that for reducing uncertainties on climate, aerosol optical depth (AOD) should be
monitored to an accuracy on the order of ±(0.03 + 10%; e.g. GCOS, 2011). The Atmospheric
Clouds and ocean Ecosystems (ACE) white paper called for an accuracy of ±(0.02 + 10%) [*Starr*
*et al.,* 2010]. To meet such tight criteria, all aspects of traditional aerosol remote sensing
methods require re-examination with the objective to reduce uncertainties in the final retrieval,
and to assure continuity as the aerosol climate data record is passed from one sensor to the next.

In this paper we focus on gaseous absorption. Aerosol retrieval algorithms (*Vermote et*

*al*., 1997) tend to use satellite observations taken in wavelength regions where gas absorptions
are small. However, while gas absorption is small in these "window" bands, it is not zero. For
example, for the 20 nm-wide Moderate Resolution Imaging Spectroradiometer (MODIS) band
near 0.55 µm, in the middle of the Chappius region, there is absorption due to ozone. For a US
1976 Standard Atmosphere (US76, 1976), with total column ozone of 344 Dobson Units (DU),
the gas absorption optical depth ($\tau^{GAS}$) is about 0.03 in this band. This is of similar magnitude as
pristine AOD (~0.05), and equal to the required measurement accuracy (GCOS; 2011). Water
vapor, measured as precipitable water vapor (PW or *w*), absorbs as well and introduces even
greater uncertainty. For example, the *w* of the US76 standard atmosphere is a modest 1.4 cm,
which translates to $\tau^{GAS}$ of about 0.025 in the MODIS 2.11 µm band or a $\tau^{GAS}$ of 0.05 for a
similar-wavelength Visible Infrared Imaging Radiometer Suite (VIIRS) band centered near 2.25





μm. The major difficulty with ozone and water vapor is that the total column burden of these
gases varies spatially and temporally over the globe [*Hegglin et al.*, 2014]. Other trace gases,
including carbon dioxide and methane, also absorb shortwave radiation in wavelength specific
regions. While these gases are more evenly distributed (well-mixed) throughout the globe, failing
to correct for their absorption also would lead to errors in aerosol retrieval.

Different aerosol retrieval algorithms respond to the challenge of gaseous correction

differently.  Some include all gaseous absorbers and account for the variability of water vapor
and ozone [*Levy et al.*, 2013; 2015], while others use a fixed ozone concentration [e.g Thomas et
al., 2010; Sayer et al., 2012], and others correct for some gases, but consider the effect of other
gases to be too small to bother with [MISR ATBD].  Few include methane [Levy et al., 2013;
2015]. How does a less complete gaseous correction scheme affect the global retrieval of AOD?
How sensitive are gaseous absorption schemes to slight shifts in spectral bands from instrument
to instrument?  While all operational aerosol retrieval algorithms employ gaseous correction
schemes in their retrieval and describe these schemes, more or less, within the "gray literature"
of internal documentation, there are few recent articles in the peer-reviewed literature that openly
describe the process and quantify the impact of the subtle choices made during algorithm
development.

In this paper we re-examine gaseous correction as it is applied in the traditional MODIS

Dark Target (DT) aerosol retrieval [*Levy et al.*, 2013], and as that retrieval algorithm is ported to
the new VIIRS data [*Levy et al.*, 2015].  In Section 2 we discuss the absorption of radiation by
atmospheric gases within the MODIS and VIIRS bands used for the DT aerosol retrieval. We
introduce the relationship of gas abundance to its transmittance spectra, which is the theoretical
basis for gas corrections in DT AOD retrievals. The atmospheric correction methodology is



detailed in Section 3. The impact of the updated atmospheric gas corrections applied to the
Collection 6 MODIS AOD is also briefed in Section 3. In Section 4 we discuss the importance of
accurate atmospheric gas corrections in context of DT AOD retrievals from the VIIRS
instrument. The study is summarized and concluded in Section 5.

## 2. The DT approach to aerosol retrieval and gas correction

### 2.1 The DT aerosol algorithm and wavelength bands

As explained in detail by Levy et al. (2013; 2015) and references therein, the dark-target
(DT) aerosol algorithm uses seven channels (or bands) covering the solar reflective spectral
region from blue through the shortwave infrared (SWIR) to characterize aerosols, clouds and
Earth's surface. These bands were specifically chosen to correspond to the spectral window
regions of minimal gas absorption. On MODIS, these bands include B1, B2, B3, B4, B5, B6 and
B7, which are each 20-50 nm in width and centered near 0.65, 0.86, 0.47, 0.55, 1.24, 1.63 and
2.11 μm, respectively.  On VIIRS, the DT algorithm uses bands M3, M4, M5, M7, M8, M10 and
M11, which are the "moderate resolution" or M-bands that are centered near 0.48, 0.55, 0.67,
0.86, 1.24, 1.60 and 2.26 μm, respectively.
The DT algorithm is actually two algorithms, one applied to MODIS- or VIIRS-measured
reflectance over land surfaces and the other to measured reflectance over ocean (Levy et al.,
2013; 2015). Both the land/ocean algorithms employ a single atmospheric gas correction method
before any retrieval is performed. DT uses a LUT approach in which atmospherically corrected
observed top-of-atmosphere (TOA) reflectance (as measured by the satellite) is compared with
simulated reflectance. The simulations are calculated by radiative transfer codes, and account for



multiple scattering and absorption effects of a combined surface (land or water), molecular
(Rayleigh), and aerosol scene, but do not account for gaseous absorption. These simulations also
account for the angular dependence of the scattered radiation, through use of a pseudo-spherical
approximation (e.g. Ahmad and Fraser, 1991). The DT retrieval operates on regions of pixels for
which cloud pixels, glint pixels, and other unsuitable pixels have been masked out. Thus, the DT
aerosol retrieval is performed for cloud-free sky, and assumptions have been made for the
surface reflectance properties and atmospheric constituents.     The LUT is interpolated as a
function of observing geometry (solar and view zenith and azimuth angles), and then searched to
determine which aerosol conditions provide spectral reflectance that best "matches" the spectral
reflectance observed by the satellite. The reported solution (retrieved spectral AOD) is some
function of the solutions that meet sufficient criteria for matching the observations. For the DT
algorithm, expected uncertainty for retrieved AOD at 0.55 μm (as compared to global network of
sun photometers) is ±(0.05 + 15%) over land, and ±(0.03 + 10%) over ocean [*Levy et al.*, 2013].

These LUTs are created as if the atmosphere is composed only of aerosol and scattering

(Rayleigh) molecules. The gas absorption is assumed to be zero. This is because of the large
spatial/seasonal variability of two of the primary absorbers: ozone and water vapor.  Ozone can
range from 100 to 500 DU around the globe   [*Hegglin et al.* 2014] and water vapor varies by an
order of magnitude from the wet tropics to the dry poles.  It would be cumbersome and
computationally inefficient to add two or more new indices to the LUT and cover the dynamic
range of each gas in the LUT calculation.

While gas absorption in these window bands may be small, they are not zero, as

described above.  Figure 1 shows the TOA transmission spectra (black lines) in the 0.4 – 2.5 μm



spectral range in the presence of major gases, including $H_2O$, $O_3$, $CO_2$, $CH_4$, $O_2$, $N_2O$, and CO.
The transmission spectra of each gas were calculated using the Line-by-Line Radiative Transfer
Model (LBLRTM) code [*Clough et al.*, 1992, 2005] for a nadir viewing geometry and for the US
1976 Standard Atmosphere (US76; 1976).  A transmittance of 1.0 indicates that the atmosphere
is transparent to incoming solar radiation (insolation) i.e. it is not absorbed in the atmosphere.
Overlaid on Fig. 1 are the spectral response functions of the seven MODIS channels (blue
curves) and seven VIIRS channels (red curves) used in the DT retrievals. As can be seen from
Fig. 1, depending on the wavelength, the atmosphere can be totally transparent to a certain gas
and partially opaque to another. For example, in the MODIS 0.62-0.67 μm band (B1), $H_2O$, $O_3$,
and $O_2$, absorb radiation while $CO_2$, $N_2O$, CO, and $CH_4$ do not. In the $1.230 - 1.250$ μm band
(B5), $O_2$, $H_2O$ and $CO_2$ are major absorbers while other gases are not. Absorption bands of the
major atmospheric gases are listed in Table 1.

Note that there are also wavelength regions that are nearly opaque because of gas

absorption. For example, Fig. 1 shows the well-known water vapor absorption within the
wavelength region near 1.38 μm. Because of the strong absorption, the 1.38 μm band cannot be
used for aerosol retrieval.  Yet, this band is very useful for detecting cirrus clouds that would
otherwise contaminate a cloud-free aerosol retrieval (Gao et al., 2002). This special case of using
absorption information is not discussed further in this paper.
**2.2 Derivation of a gas absorption correction**

Because the LUT is calculated without gas absorption, an alternative technique must be

substituted to account for the effect of the gases in each wavelength. If not, then when the
algorithm attempts to match the measured TOA reflectances to the LUT-calculated reflectances



the LUT values will be brighter than the measured values for the same amount of aerosol. In the
most straightforward sense retrieved AOD, dominated by scattering, will be systematically too
low because the retrieval will be searching for a less bright TOA reflectance in the LUT, with
less aerosol, to match the observed values. The algorithm deals with this mismatch between
measured and LUT reflectance caused by the missing gas absorption in the LUT values by
adjusting *the measured* TOA reflectances in each wavelength band, in effect brightening the
measurements to better match the values in the LUT.
Figure 1 shows that five gases ($H_2O$, $O_3$, $CO_2$, $O_2$ and $CH_4$) have absorption lines that fall
within the wavelength bands used for the DT aerosol retrieval. Because each window band spans
tens of nanometers, every DT channel is affected by at least one gas where the transmittance is
less than 1.0.
We have introduced two measures, gas opacity and transmissivity corresponding to the
gas absorption optical depth and transmittance. The two parameters are related via,

$$T_\lambda^i = \exp\left(-G^i \tau_\lambda^i\right) \qquad \ldots\ldots\ldots (1)$$


where $T_\lambda^i$ is the downward transmittance for a particular wavelength band or λ, and for a
particular absorbing gas "$i$", and where $\tau_\lambda^i$ is the gas optical depth designated for the particular
gas and wavelength and $G^i$ is the airmass factor (slant path through the atmosphere) for gas $i$.
Equation (1) shows that transmission of light is a function of the atmospheric path length ($G^i$)
and the gas optical depth ($\tau_\lambda^i$), and that transmissivity decreases with increasing air mass and
increasing gas concentration.

**2.2.1 Gas optical depth**
The gas optical depth, $\tau_\lambda^i$, represents the spectral integral over the wavelength band, and
if the gas concentration was uniform along the path (column), then $\tau_\lambda^i$ would be directly
proportional to the loading of gas $i$ in the column. Some gases are indeed well-mixed in the
atmosphere, but water vapor and ozone are not.  These important absorbers exhibit distinctive
vertical profiles, as will be discussed in Section 3.1. Note that each individual gas has its own
particular absorption efficiency based on its characteristic absorption cross section, and that for
the same column concentrations $\tau_\lambda^i$ will be different for different gases. In the absence of a long
slant path, and for small gas optical depths ($\tau_\lambda^i \ll 1.0$), transmission can be estimated by $T_\lambda^i \sim 1-$
$G^i \tau_\lambda^i$.

**2.2.2 Airmass factor**
The airmass factor, G can be approximated as G=1/cosZ where Z is the zenith angle, for a
homogenous (exponential decay) atmosphere, and for small values (near nadir) of a zenith angles
Z.  This is the flat earth approximation. As Z increases beyond 60°, the air mass factor is more
accurately described by spherical shell geometry towards the horizon [Gueymard, 1995], i.e.:

$$G = \sqrt{(r\cos Z)^2 + 2r + 1} - r\cos Z \qquad \ldots\ldots\ldots (2)$$

where, $r = R_E / H_{atm}$ ; $R_E$ = radius of Earth (6371km) and $H_{atm}$ = effective scale height of the
atmosphere (approx 9km). This expression accounts for Earth's sphericity and atmospheric
refraction. Differences in computing G are small for Z<70°, but increase to 10% as Z = 84° (the
maximum zenith angle allowed within the DT algorithm).
Yet, there are complications. When atmospheric constituents are well-mixed and their
concentrations are nearly proportional to altitude within the atmosphere, Eq (2) is sufficient.
However, water vapor (concentrated near the surface) and ozone (concentrated in the





stratosphere) are not well-mixed in the vertical, having different scale heights. In this layered
situation (rather than continuous), there are empirical formulas (e.g. *Kasten and Young*, 1989)
that provide slight improvements to the calculation of G assuming spherical geometry.  For
example, *Gueymard*, [1995] derived the empirical formula

$$G^i = (cosZ + a_{i,1}Z^{a_{i,2}} * (a_{i,3} - Z)^{a_{i,4}})^{-1} \quad ......... (3)$$

where $a_{i,j}$ are coefficients ($j$=1,4) for gas type $i$. Thus, $G^i$ varies with gas type and specific profile
within the atmosphere. The values of coefficients $a_{i,j}$ can be found in Table 4.1 of *Gueymard*,
[1995].

As long as the total gas optical depth is small ($\sum_i \tau_\lambda^i \ll 1.0$), the total transmission of all

trace gases is well-approximated by the product of each individual gas (i.e).:

$$T_\lambda^{GAS} = \prod_i T_\lambda^i = \exp\left(\sum_i -G^i\tau_\lambda^i\right) \quad ......... (4a)$$

The total gas transmissivity defined in Equation (4a) for each wavelength band quantifies the
degree to which the measured reflectance will be diminished due to gaseous absorption. In order
to match the measured reflectances to those calculated for the LUT, these diminished
reflectances have to be "corrected" or brightened.  This correction factor is simply the inverse
transmissivity, $\tilde{T}$ ,

$$\tilde{T}_\lambda^{GAS} = 1/T_\lambda^{GAS} = \exp\left(\sum_i G^i\tau_\lambda^i\right), \quad ......... (4b)$$

which when multiplied with the measured reflectance restores the amount of light absorbed by
gases along the one-way path of transmission. Or, given a measured radiance, $L_M$, the corrected
(brightened) radiance $L$, is simply, $L = L_M * \tilde{T}$.





When observing from a ground-based sun photometer (e.g. AERONET), the correction is
straightforward, because the path of transmission traverses the depth of the atmosphere only
once. The problem is more complicated for satellite remote sensing, because a satellite measures
radiation that has traveled downwards through the atmosphere and then back up to space.  We
have to calculate a two-way correction factor and *G* must account for the *Z* angles of both
downward (the solar zenith angle) and upward paths (view zenith). As Z gets large, the vertical
profile of the gas (layering) becomes more important.
There are two parameters determining the transmission, $T_\lambda^{GAS}$, and therefore the
correction factor, $\tilde{T}_\lambda^{GAS}$, and these are $G^i$ and $\tau_\lambda^i$. The goal, then, is to parameterize equation 4(a)
or 4(b) i.e. the relationship between atmospheric transmission of gas and $G^i \tau_\lambda^i$ ; taking into
consideration the varying gas concentrations and their vertical profiles through the atmosphere,
around the globe.  Furthermore, the parameterization will be developed to link $\tilde{T}_\lambda^{GAS}$ directly to
column measures of the gases instead of to the optical depth.  This allows the algorithm to
bypass calculations of optical depth from inputs of precipitable water vapor (*w* in cm) and ozone
(*O* in Du), and instead use the inputs directly.

**3. Use of LBLRTM to derive gas absorption parameterization**

To develop an empirical relationship between atmospheric gas transmission, the airmass
factor ($G^i$) and its optical depth ($\tau_\lambda^i$), we require a radiative transfer (RT) code that can
accurately simulate the gaseous absorption and transmission process in the atmosphere. Among
other things, the RT code requires these two pieces of information: (a) the absorption cross-
sections and concentration of gas constituent in spectral bands of interest  and (b) accurate high-





resolution information of the absorption spectra of the relevant gases. The MODIS and/or VIIRS
channels widths are on order of 20-50 nm. We require a high-resolution database to capture the
fine absorption lines within these bandwidths.  To address (a) and (b), we use the Line-By-Line
Radiative Transfer Model (LBLRTM) to parameterize equations 4(a) and 4(b) instead of a
MODTRAN based RT code. The following section provides details of LBLRTM.

3.1 LBLRTM description
The Line-By-Line Radiative Transfer Model (LBLRTM) is known to be an accurate and
flexible radiative transfer model that can be used over the full spectral range from ultraviolet to
microwave [Clough et al., 2005].  It uses the High-resolution TRANsmission (HITRAN)
molecular absorption database [*Rothman et al.*, 2009] for calculating transmittance and radiance
of molecular species. The HITRAN2008 database contains over 2,713,000 lines for 39 different
molecules. The spectral resolution of the data is different in different spectral regions and for
different species [see *Rothman et al.*, 2009]. For example, for water vapor absorption in the Near
IR region, the line resolution is 0.001 cm$^{-1}$ [2.5 – 3.4 µm]. The LBLRTM has been extensively
validated against atmospheric radiance spectra [e.g. *Turner et al.*, 2003; *Shephard et al.*, 2009;
*Alvarado et al.*, 2013]. Use of the HITRAN database and other attributes of LBLRTM provide
spectral radiance calculations with accuracies that are consistent with validation data. Limiting
errors are, in general, attributable to line parameters and line shape.  Algorithmic accuracy of
LBLRTM is approximately 0.5% and is about five times less than the error associated with line
parameters [*Clough et al.*, 2005].
3.2 LBLRTM calculations for MODIS and VIIRS





The LBLRTM model was run for many scenarios representing different combinations of
gas vertical profiles, gas concentrations and air mass factors for each type of gas and each of the
wavelength bands of interest.  Transmissions of the ten important atmospheric gases, viz. $H_2O$,
$O_3$, $O_2$, $N_2O$, $NO_2$, $NO$, $SO_2$, $CO_2$, $CO$, and $CH_4$, that affect either the MODIS or the VIIRS
spectral bands [*Levy et al*., 2013] were calculated.  The results link transmission, $T_\lambda^i$, or gas
correction factor, $\tilde{T}_\lambda^i$, to gas path length, $G^{H_2O}w$ or $G^{O_3}O$, for water vapor ( $H_2O$) and ozone ($O_3$),
respectively, where $w$ is the precipitable water vapor in cm and $O$ is ozone column loading in
DU. Values for $w$ and $O$ are input into the algorithm from ancillary data. The other gases are
considered to be well-mixed and not varying spatially or temporally, and therefore, are not
dependent on input ancillary data. The final parameterization will be curve fits through the
scatter of the model results.
As described in Section 2.2, $\tilde{T}_\lambda^{GAS}$ will be affected by the vertical distribution of the gases
in the column, especially at oblique zenith angles. To account for this effect in building the
parameterization we use 52 atmospheric profiles (personal communication, Pubu Ciren, NOAA)
that were obtained from model runs, and characterize different locations and seasons (Figure 2).
The columnar gas concentrations differ across the 52 profiles, varying by more than a factor of
ten for water vapor, and by 100% for ozone. Except for $NO_2$, which is highly variable in both
horizontal and vertical, the other trace gases tend to be well-mixed throughout the atmosphere.
Using radiative transfer calculations, Ahmad et al., (2007) show that $NO_2$ has largest impact
(1%) on TOA reflectance in the blue channels (412 and 443 nm). Other visible channels are
impacted to a lesser degree. We will use the term 'dry gas' to denote the eight gases that are
neither $H_2O$ or $O_3$, and use the US 1976 Standard Atmosphere (US 76) as a default profile.




For $H_2O$ and $O_3$, and each of their respective profiles, we use LBLRTM to calculate air
mass factors and transmissions for 10 values of viewing zenith angle, ranging from $0° − 80°$.
Transmission is integrated across the wavelength band, and weighted by relative sensor response
(RSR) (Barnes et al., 1998; Xiaoxiong et al., 2005) within the band. Because air mass factor ($G^i$)
varies with gas type (on account of the vertical profile), LBLRTM calculates $G^i$ as well as
transmission for the given column amount of gas $i$. For "dry gas", the integrated RSR weighted
transmission is converted to gas optical depth, so dry gas transmission (as a function of air mass
factors) is easily computed using Eq (1). The US 1976 Standard Atmosphere (US 76) profiles are
used to compute "dry gas" transmission for nadir view.
Figure 3 plots the relationship between absorption correction factor, $\tilde{T}_\lambda^{GAS}$, and gas path
length, $G^{H2O}w$, for $H_2O$ (panel a) and, $G^{O3}O$, for $O_3$ (panel b), for MODIS. Figure 4 plots the
same for VIIRS. These correction factors (inverse of transmission) are plotted for each window
band, for different combinations of $H_2O$ or $O_3$ concentrations ($w$ in cm or $O$ in DU) and
internally derived air mass factors ($G^i$) for the given gas type and specific vertical profile. For
water vapor (panels (a) in both figures), a near-linear dependence of $\tilde{T}_\lambda^{H_2O}$ to $G^{H2O}w$ exists in log-
log space. Water vapor, being so variable as well as concentrated near the boundary layer, cannot
be explained with a linear relationship. Even within the log-log space, there is small curvature
that requires a quadratic for the empirical fit. For ozone, however, the log of our correction factor
($\tilde{T}_\lambda^{O_3}$) is nearly linear as a function of absorption through a slant path ($G^{O3}O$). Again, note that $G^i$
is computed by LBLRTM, and represents the curvature and vertical profile of each gas type.



Equation 5 describes the quadratic empirical relationship (seen in Fig.3a and Fig 4a)
between gas transmission correction factor of water vapor ($\tilde{T}_\lambda^{H_2O}$), its concentration ($w$) and air
mass factor ($G^{H2O}$):

$$\tilde{T}_\lambda^{H_2O} = \exp\left(\exp\left(K_{1,\lambda}^{H_2O} + K_{2,\lambda}^{H_2O} \ln(G^{H_2O}w) + K_{3,\lambda}^{H_2O} (\ln(G^{H_2O}w))^2\right)\right) \quad \ldots\ldots\ldots (5)$$


and Equation 6 describes the near linear relationship for ozone (panels b in both Fig. 3
and Fig. 4).

$$\tilde{T}_\lambda^{O_3} = \exp\left(K_{1,\lambda}^{O_3} + K_{2,\lambda}^{O_3}(G^{O_3}O)\right) \quad \ldots\ldots\ldots (6)$$

"$O$" denotes ozone concentration in Eq. 6 and $G^i$ is the airmass factor for gas $i$ and is
computed using equation 3.
The regression coefficients $K_{1,\lambda}^{H_2O}, K_{1,\lambda}^{H_2O}, K_{1,\lambda}^{H_2O}$ and $K_{1,\lambda}^{O_3}, K_{2,\lambda}^{O_3}$ (the slopes and intercepts)
for $H_2O$ and $O_3$ are presented for MODIS and VIIRS in Tables 3.1 and Table 3.2. The slope and
intercepts are wavelength dependent (lines of different color on Figs. 3 and 4) and in accordance
to absorption characteristics of the gas. For example Table 2.1 shows that water vapor absorption
is highest in MODIS band 7  (B7 = 2.11 μm) and lowest in B3 (0.47 μm). Correspondingly, the
slope and intercept for the $H_2O$ regression relation (Table 3.1) indicates largest water vapor
correction in B7 and lowest in B3. Similarly, largest correction (and slope) for ozone is in
MODIS B4 (0.55 μm) and lowest in B7.




To calculate the correction factors for water vapor ($\tilde{T}_\lambda^{H_2O}$) and ozone ($\tilde{T}_\lambda^{O_3}$), Equations (5)
and (6) require information on water vapor (w) and ozone concentration (O). For the DT
algorithm, these are provided by an ancillary data set. For the current version (e.g. MODIS
Collection 6), ancillary data are acquired from National Center for Environmental Prediction
(NCEP) analysis, specifically the "PWAT" and the ozone fields from the 1° X 1° global
meteorological analysis (created every six hours – format "gdas.PGrbF00.YYMMDD.HHz").
Note that there are water vapor products derived operationally from MODIS and VIIRS data
(e.g. Gao and Goetz, 1990; Kaufman and Gao, 1992). However, the DT aerosol algorithm runs
before these other algorithms in the processing chain, causing the internally-derived water vapor
to be unavailable to the aerosol algorithm in real-time processing and thus, the reliance on
ancillary data.
In case the ancillary information is not available, the gas absorption can still be estimated.
Either a forecast field (e.g. GDAS forecast) or a "climatology" can be used. For example, if the
US76 atmosphere is assumed as the climatology for gas profiles, then $\tau^i$ for that gas is given in
Table 3.1 and 3.2. In this case, we use Equations (7) and (8) to calculate correction factors for
water vapor and ozone respectively:

$$\tilde{T}_\lambda^{H_2O} = \exp\left(G^{H_2O}\ \overline{\tau^{H_2O}}\right) \qquad ......... (7)$$

$$\tilde{T}_\lambda^{O_3} = \exp\left(G^{O_3}\ \overline{\tau^{O_3}}\right) \quad ......... (8)$$

where $\overline{\tau^{H_2O}}$ and $\overline{\tau^{O_3}}$ are the climatological mean values of gas optical depth for water vapor
and ozone, respectively.
$\tilde{T}_\lambda^{Dry\ Gas}$ is the correction factor due to dry gas, which includes $CO_2$, $CO$, $N_2O$, $NO_2$, $NO$,
$CH_4$, $O_2$, $SO_2$, other trace gases. Since the gases are generally well-mixed throughout the entire
atmosphere and do not experience day to day changes, we only consider the climatological mean



of the total optical depth of the combined dry gases, and compute its transmittance factor as
follows:

$$\tilde{T}_\lambda^{Dry\ Gas} = \exp\left(G^i \overline{\tau^{Dry\ Gas}}\right) \quad \dots\dots\dots (9)$$


Fig. 5 presents the gas optical depth for a US76 atmosphere, for the MODIS bands and
corresponding VIIRS bands. In some cases, (e.g. B4 vs. M5) the differences are small.  In other
cases (e.g. B5 vs. M8), the total optical depth may be similar, but the relative contribution
between different gases different. Finally, in at least one set of bands (B7 vs. M11), both the total
optical depth and the relative contributions between gases is very different.  The US76 is a case
of a small amount of water vapor ($w$=1.4 cm), but one can see how quadrupling the $w$ (e.g. as in
a tropical atmosphere) would greatly change the relative correction needed for B7 vs. M11, or
even B1 vs. M5.

**3.3 Application within the DT algorithm.**
Whether using "climatology" for water vapor and ozone columns, or using the estimates
from a meteorological assimilation system (e.g. GDAS for the current DT algorithm), we need to
correct for the combined absorption of all gases. The total gas absorption correction term, $\tilde{T}_\lambda^{gas}$,
is the product of individual gas corrections, that is

$$\tilde{T}_\lambda^{gas} = \tilde{T}_\lambda^{H_2O} \tilde{T}_\lambda^{O_3} \tilde{T}_\lambda^{Dry\ Gas} \dots\dots\dots (10)$$

The MODIS DT aerosol retrieval algorithm ingests calibrated and geolocated MODIS-
measured reflectance data, known as the Level 1B (L1B) product. The corresponding VIIRS DT
algorithm ingests a similar VIIRS-measured product. This measured reflectance, $(\rho_\lambda^{L1B})$, is





corrected for atmospheric water vapor, ozone and dry gas, using the correction factors derived
above for each wavelength band:

$$\rho_\lambda = \tilde{T}_\lambda^{gas} \rho_\lambda^{L1B} \ldots \ldots \ldots (11)$$

where $\rho_\lambda$ is the corrected or brightened reflectance that will now match the calculated TOA
reflectances of the LUT, as described in Section 2.2. Note that this spectral reflectance $\rho_\lambda$,
represents the combination of Rayleigh (molecular scattering), plus aerosol in the atmosphere. It
also includes contributions from Earth's surface (land or water).

The gas-absorption correction methodology is the same whether performed for MODIS

or VIIRS. In fact, the equations (Eqs 5-11) have remained the same throughout all versions of
the DT algorithm. As our ability to characterize absorption lines as well as the spectral response
of the sensor has improved, it is the coefficients of the equations that have evolved. When the
DT algorithm was updated from Collection 5 (C5) to Collection 6 (C6), the underlying gas
absorption corrections became more sophisticated (Levy et al., 2013). This is represented in
Table 4. The primary differences between C5 and C6 are that HITRAN database in LBLRTM is
used in C6 instead of the MODTRAN parameterization available in 6S that was used in C5, and
that additional "dry" gases have been included in C6's correction. These changes made a
difference. The latest version of aerosol data from DT is Collection 6.1 that uses the same gas
absorption corrections as C6. As the DT algorithm is ported from MODIS to VIIRS data, the
quality of gas correction will also make a difference.
**4 Impact of new gas coefficients**

The DT retrieval is based on a LUT approach wherein the measured and modeled spectral

reflectance is matched for inversion. Any change affecting the calculation of gas-corrected



spectral reflectance will subsequently affect the retrieved AOD. *Levy et al.*, [2013] showed the
impact of using the updated atmospheric corrections on MODIS C6 AOD retrievals. This led to
higher AODs globally. Over land (ocean), the 0.55 μm global mean AOD differed by ~0.02
(0.007). The large (>0.02 regionally) change over land was primarily due to a larger gas
correction in the 1.24 μm MODIS B5 band (see *Levy et al.*, 2013; Fig. A2), which in turn
increased the reflectance in B5, and subsequent estimate of the NDVI in the SWIR channels (B5
vs. B7) used to estimate surface reflectance in other bands (Levy et al., 2010). The stronger gas
correction in B5 came from including the $O_2$ absorption, which had not been accounted for in C5
(see Table 2.1). Interestingly, Levy et al. [2013] noted that while the overall correction in B7
(2.11 μm) remained similar, the relative weightings of "dry gas" and $H_2O$ was revised.
Even though MODIS and VIIRS instruments have similar channels, the MODIS gas
correction coefficients cannot be applied to aerosol retrievals from VIIRS observations. The
slight differences in the bandwidth and channel's central wavelengths (See Fig. 5) will
compromise the accuracy of aerosol retrievals. For example, as compared with MODIS B7 (2.11
μm), the VIIRS M11 (2.25 μm) band has less absorption from $H_2O$. However, MODIS B7 lies in
a $CO_2$ absorption band, while VIIRS M11 lies in a region of $CH_4$ absorption. Although the $CH_4$
optical depth in VIIRS M11 is small (~0.03), it will affect the dark-target retrievals in the same
way as $O_2$ inclusion affected C6 retrievals (when compared to C5).
As a perturbation experiment we intentionally apply the MODIS gas corrections to the
VIIRS observations, even though we know this to be incorrect. Figure 6a plots the spatial
distribution of spectral TOA reflectance after applying VIIRS-appropriate gas corrections. It
shows the mean monthly TOA reflectance for VIIRS. Figure 6b are the reflectance differences





between applying VIIRS-appropriate gas corrections and MODIS gas corrections to VIIRS
observations.    From top to bottom, we find a mean difference of 0%, -0.5%, -6.6%, -2.7%, -
1.5%, 3.2% and 5.3% respectively in VIIRS channels M3, M4, M5, M7, M8, M10, M11.
Looking back at Fig. 5, one can see that that for example, by using proper M5 assumptions
instead of the B1 MODIS assumptions, we now apply only about half the correction as before,
resulting in a 6.6% reduction of reflectance. Channel M7, with about 50% less water vapor
correction (see Fig. 5), results in 2.7% lower reflectance. Larger gas corrections owing to $CO_2$
absorption in M10 and $CH_4$ absorption in M11 (Fig. 5), result in positive bias in M10 and M11
reflectance values globally.

Now, we continue the perturbation experiment and test the impact of slight differences in

the band positioning between MODIS and VIIRS on AOD retrieval by performing two sets of
retrievals. The first set (a) is if we applied appropriate VIIRS band corrections, while the second
(b) is as if we had simply (naively) applied MODIS (C6) coefficients to VIIRS data. Figure 7
shows the AOD retrieved from these two cases (panels a and b) for an entire month (July 2013)
of VIIRS data. While general AOD spatial patterns are in agreement, panel (c) shows differences
in AOD of up to 0.07 between the two retrievals. Clearly, naively applied MODIS gas
corrections to VIIRS data, would lead to a global mean AOD underestimate of ~0.01 for July
2013.   While these differences are within the global uncertainties for AOD (e.g. GCOS), the
regional differences can be much larger.

Although once considered to be trivial in magnitude, accurate atmospheric gas

corrections have become more important as we strive towards better accuracies in AOD products
and towards a seamless climate data record. It is noteworthy that the gas absorption spectra of
Figure 1 have been updated several times in recent years [Alvarado et al. 2012] as the scientific





community continues to engage in study of gas absorption lines with improved instrumentation
and gas spectroscopic measurements. Changing gas absorption spectra will affect the channels
designed for new remote sensing instruments and in understanding how these lines might affect
the retrieval of proposed geo-physical products. Every instrument design involves
characterization of channel bandwidths and the spectral response functions of the instrument's
channels. This aptly calls for updates in modeling the absorption by gases in the channels used
for aerosol retrievals. For the MODIS Collection 6 AOD product, the team switched from using
a MODTRAN gas spectroscopic database to the HITRAN spectroscopic database and found
differences.
**5. Summary and Conclusions**

Performing aerosol optical depth retrieval, from satellite measurements, requires

extracting the aerosol signal from the total radiance measured by the sensor at the top-of-
atmosphere. The total radiance includes signal from the underlying surface and from atmospheric
constituents such as gases, clouds and aerosols. In this paper, we have described the physics and
methodology employed by the Dark-Target aerosol retrieval algorithm for atmospheric gas
correction of the cloud-free radiance measurements from the MODIS and VIIRS sensors. We
have shown that the empirical correction applied to one sensor (MODIS) cannot be applied to
another sensor (VIIRS) even when the channels of the two sensors may be similar. For a specific
month of VIIRS observations (July, 2013), not accounting for the sensor's bandwidth and
positioning of its central wavelength in the electro-magnetic spectrum, can result in an AOD
retrieval bias of about 0.01 (global average) and up to 0.07 at regional scales.





Water vapor, ozone and carbon dioxide are the major absorbers of solar radiation.
Historically, they have been accounted for in atmospheric gas corrections by aerosol retrieval
algorithms. However, until recently, standard routine algorithms (e.g. the DT algorithm used on
MODIS) did not consider other gases.  For example, oxygen with a gas optical depth of about
0.016 is important in the MODIS Band 5 (1.24 μm) [Levy et al., 2013].  Methane is an important
absorber in band M11 (2.25 μm) of VIIRS with an optical depth of ~0.05. Starting with MODIS
Collection 6, and the DT algorithm ported to VIIRS, seven additional atmospheric gases [CO,
$N_2O$, $NO_2$, NO, $CH_4$, $O_2$, $SO_2$] are addressed by the gas correction in these DT algorithms.
For the 'dry gas' component, the DT gas correction assumes a homogeneous global
distribution spatially and a US76 type of vertical distribution for the eight gases. Carbon dioxide,
oxygen, nitrous oxide and methane are major absorbers in our 'dry gas' category. Except for
$NO_2$, which is highly variable in both horizontal and vertical, the other gases tend to be well-
mixed throughout the atmosphere. Spatial variability of well-mixed gases is typically around
10%, mostly latitudinal and is smaller than seasonal variability (e.g. see methane maps here:
http://www.temis.nl/climate/methane.html). For nadir view, 10% error due to spatial variability
will only introduce an error of 0.005 in the methane correction (optical depth ~0.05 in VIIRS
channel M11). For now, this is a small uncertainty in the overall retrieval.  However, as
requirements for aerosol retrieval accuracies tighten, even these well-mixed dry gases will
require removal of any seasonal and regional biases by using ancillary measurements of these
gases or at least seasonal global climatology of gas optical depths, instead of a single
climatological value for the entire globe.
Since the DT algorithm corrects for $H_2O$ and $O_3$ using ancillary data at every 1˚ X 1˚ grid
box, spatial and seasonal variability of these gases is being accounted for. However, the ancillary



data has its own uncertainties that propagate into the gas correction and aerosol retrieval. The
Dark-Target team is working towards estimating the error in per-pixel AOD retrievals introduced
from several error sources including the errors in $H_2O$ and $O_3$ ancillary data (GDAS) used for
atmospheric gas corrections.  Preliminary analysis suggests (not shown here) that gas corrections
errors, stemming from considering 20% errors in ancillary data, are much smaller (more than an
order of magnitude) than errors from surface albedo uncertainty, aerosol model selection, spatial
heterogeneity in a scene, calibration and cloud contamination errors. This is work in progress and
subject to future publication.

The VIIRS instrument onboard Suomi-NPP is a follow-on of the MODIS instrument on

Terra and Aqua satellites. While the Dark-Target team strives to create a seamless climate data
record (CDR) of AOD from MODIS and VIIRS, it requires a consistency in AOD retrieval of
about 0.02. Any compromise with the accuracy of AOD retrieved from either sensor will impact
the CDR consistency requirement. To strive toward these requirements, we cannot ignore quality
atmospheric gas corrections in AOD retrievals and we will update the gas correction factors for
each instrument as the community updates the gas absorption database.

As we move into an era of new aerosol missions, revisiting and updating atmospheric

corrections in state-of-art algorithm becomes as important as improving upon other factors (e.g.
better surface characterization, cloud clearing, aerosol properties etc.) that affect the AOD
retrieval. The dark-target algorithm software has now been generalized to retrieve AOD from
sensors other than MODIS and VIIRS. It will be necessary to accurately characterize gases from
such current and future instruments as Himawari, GOES-R, etc.







**Acknowledgements**
This research work is funded under NASA's grants for MODIS and VIIRS Dark Target
aerosol retrieval for the MODIS science team. We are thankful to Matthew J. Alvarado (from
Atmospheric and Environmental Research) for promptly helping with all our queries related to
the LBLRTM. We thank Pubu Ciren for providing us with the atmospheric profiles for gases and
for knowledge transfer on its use in LBLRTM for calculating NOAA VIIRS atmospheric gas
corrections in aerosol retreivals




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

**Figures**








## Figure 1

The TOA Transmission spectra (black) of the major atmospheric gases in the Visible and Near Infrared part of electromagnetic spectrum (400 – 2500 nm). The Line-by-line radiative transfer model (LBLRTM) was used to calculate these gas spectra for a nadir viewing geometry and the 1976 US Standard atmosphere. The spectral response functions of MODIS channels B1-B7 (blue curves) and seven VIIRS channels (red curves) are overlaid for visualizing their positioning in atmospheric 'window' region where gas absorption effect is minimal







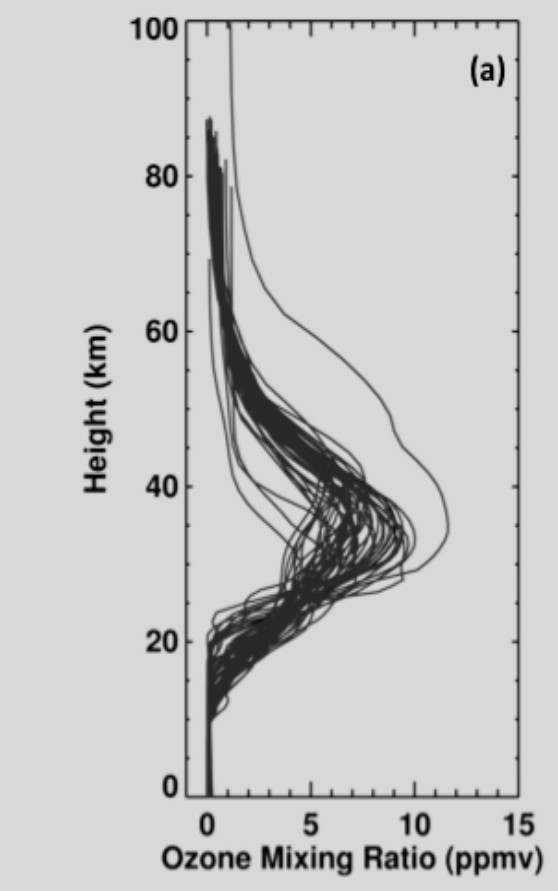
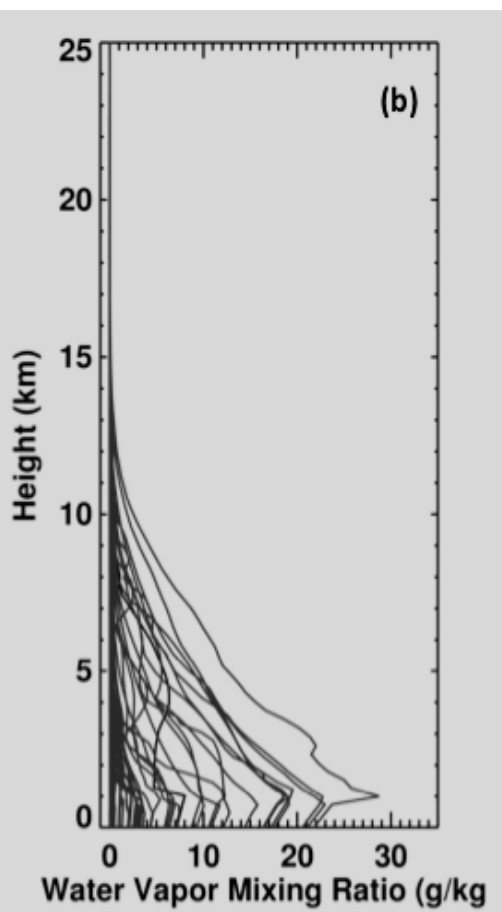

Figure 2: 52 different ECMWF profiles for (a) water vapor and (b) ozone used in the Line-by-line radiative transfer model to calculate the respective gas transmittance.







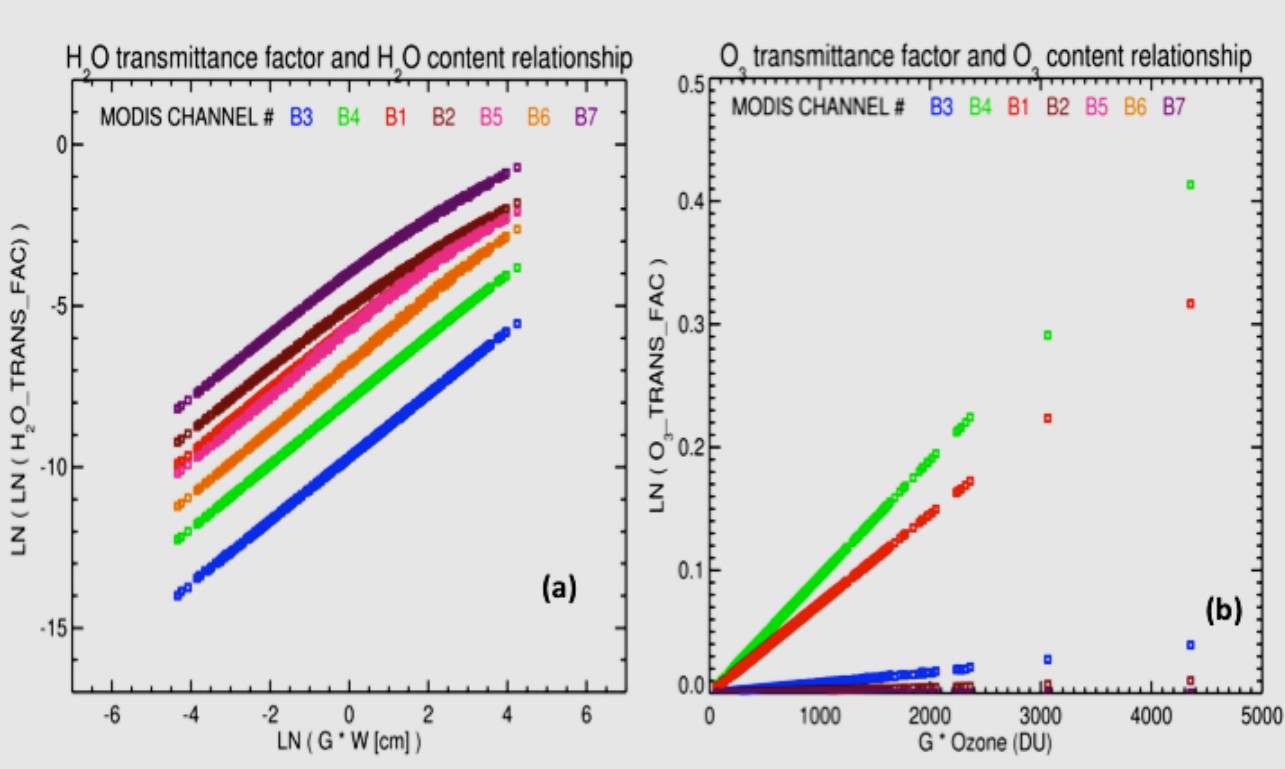

Figure 3: Relationship between Gas Transmittance factor and Gas Content in the MODIS channels B1 − B7 : (a) For $H_2O$ and (b) for $O_3$. Gas content is scaled by the airmass factor [G]







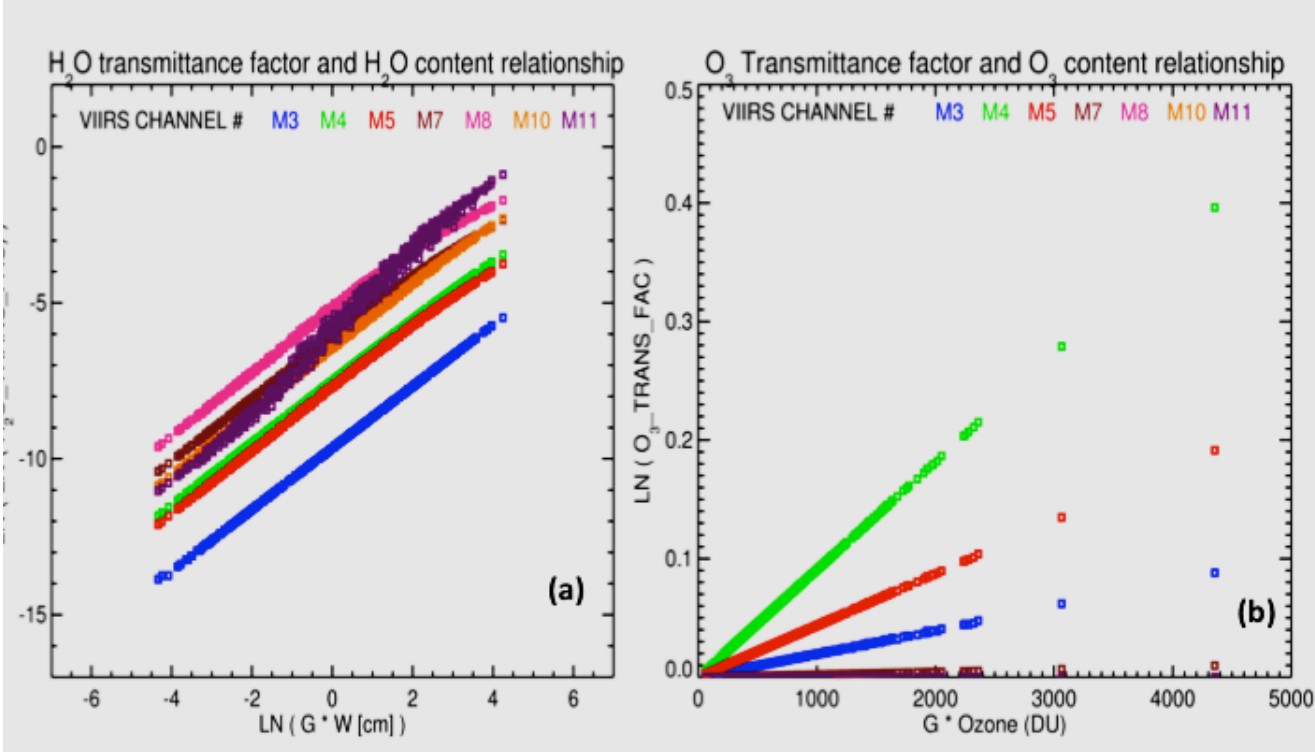

Figure 4: Relationship between Gas Transmittance factor and Gas Content in the seven VIIRS channels: (a) For $H_2O$ and (b) for $O_3$ Gas content is scaled by the airmass factor [G]






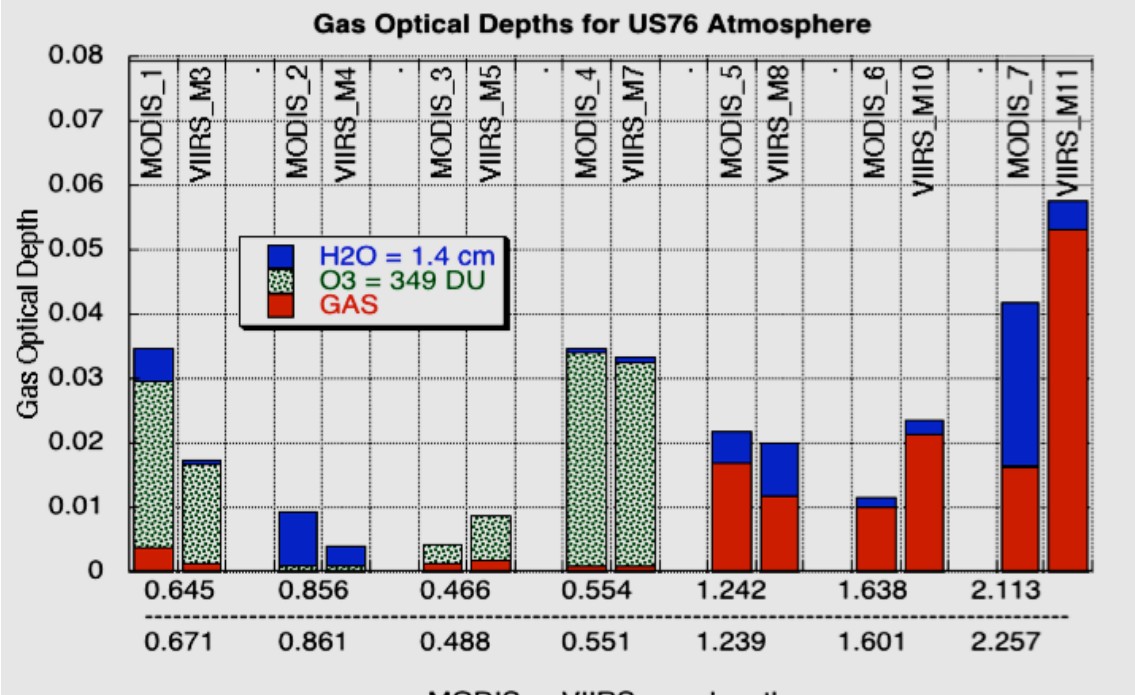

Figure 5 : Comparison of gas optical depths calculated for US 1976 standard atmosphere using MODIS C6 and VIIRS gas correction coefficients. Different colors represent constituent gases ($H_2O$ = blue, $O_3$ = green hatched, 'dry' gas = red . Large differences in gas optical depths are seen in MODIS Channels 1, 2,6 & 7.





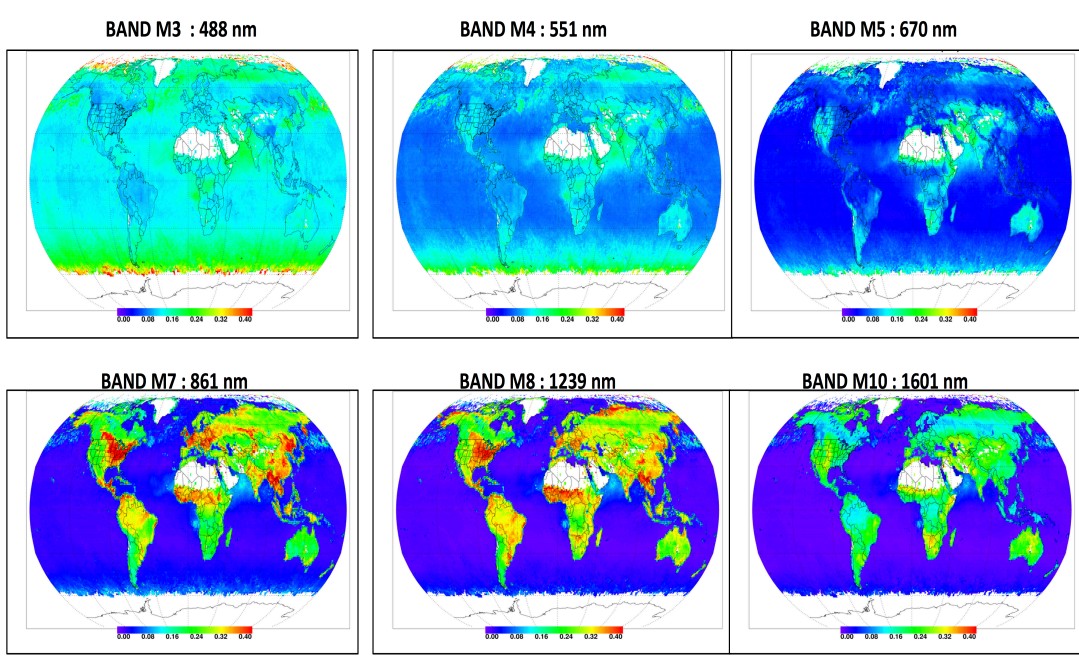

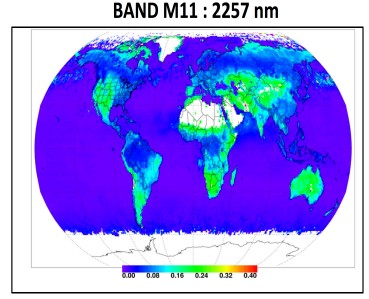

Figure 6a : This figure shows the spatial distribution of gridded L2 reflectance in 7 VIIRS channels [i.e. M3, M4, M5, M7, M8., M10, M11] for July 2013





## Difference between VIIRS – C6 Gas Reflectance

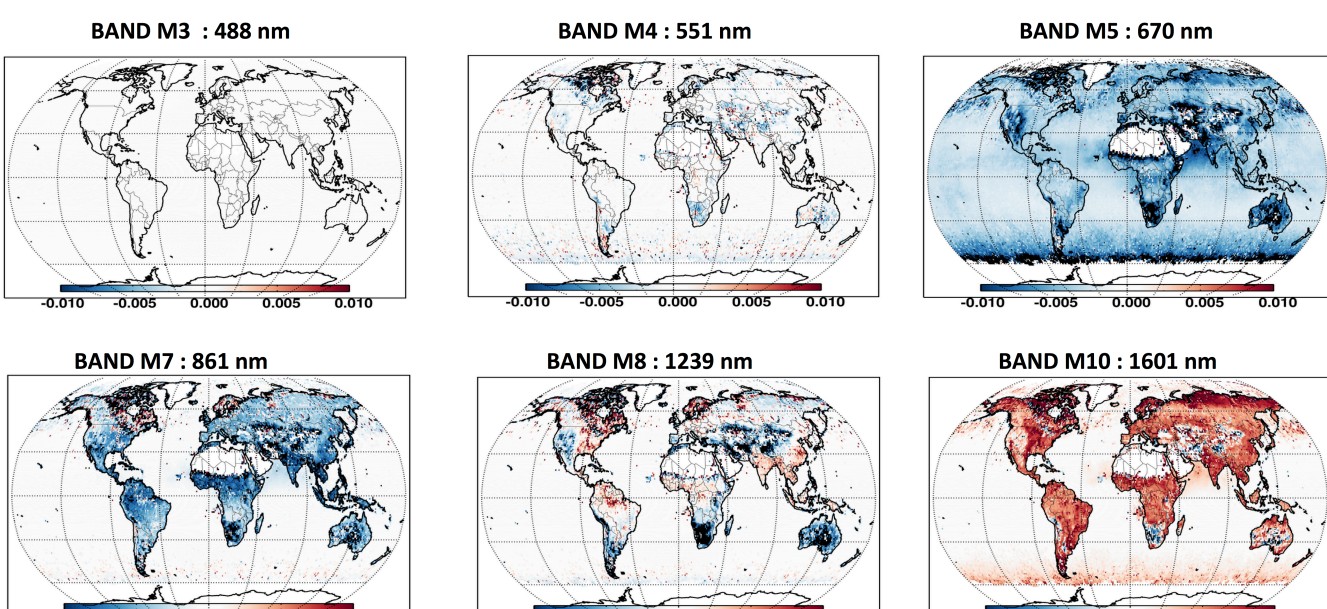

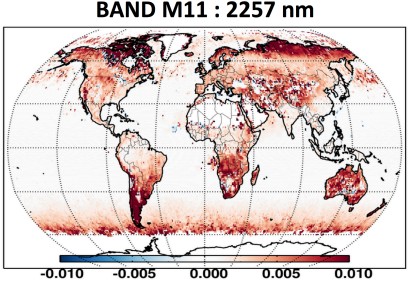

**Figure 6b** : This figure shows the spatial distribution of the difference between VIIRS reflectance obtained by applying VIIRS gas correction coefficients and MODIS C6 gas correction coefficients in 7 VIIRS channels [i.e. M3, M4, M5, M7, M8., M10, M11] for July 2013. This figure demonstrates the impact of using MODIS gas correction on VIIRS reflectance used for retrieving aerosol optical depth.





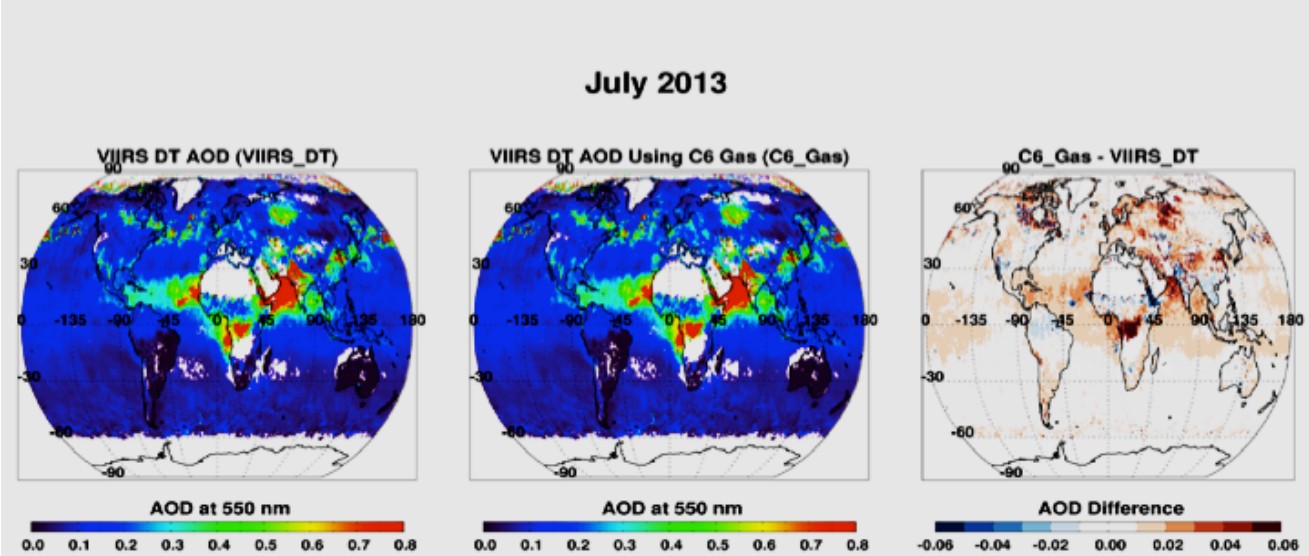

**Figure 7**: Impact of updated atmospheric corrections on VIIRS AOD (550 nm) retrieval. All things being equal, using C6 aerosol DT retrieval algorithm (a) is AOD using atmospheric coefficients calculated for VIIRS bands and (b) is AOD using C6 atmospheric corrections (c) is the difference between (b) and (a). The global mean AOD differs by ~0.012 over land and by ~0.004 over ocean. Difference are larger than these mean values regionally but <0.08. Differences are mostly positive (reds) except in some desert / bright regions where some negative differences appear.




**Tables**

**Table 1 : Absorption bands of atmospheric gases in visible and near-IR region**

| Major Atmospheric Gas | Center Wavelengths (µm) |
|---|---|
| $H_2O$ | visible, 0.72, 0.82, 0.94, 1.1 1.38, 1.87, 2.7 |
| $CO_2$ | 1.4, 1.6, 2.0, 2.7, 4.3 |
| $O_3$ | visible (0.45 - 0.75) |
| $O_2$ | 0.63, 0.69, 0.76, 1.06, 1.27, 1.58 |
| $N_2O$ | 2.87, 4.06, 4.5 |
| $CH_4$ | 1.66, 2.2, 3.3 |
| CO | 2.34, 4.67 |
| $NO_2$ | visible |





**Table 2.1 : Optical depth of major atmospheric gases in 7 MODIS channels.**


| Channel | B3 | B4 | B1 | B2 | B5 | B6 | B7 |
|---|---|---|---|---|---|---|---|
| Wavelength (μm) | 0.466 | 0.553 | 0.645 | 0.856 | 1.242 | 1.638 | 2.113 |
| Gas | | | | | | | |
| H2O | 0.0001 | 0.0005 | 0.0055 | 0.0086 | 0.005 | 0.0017 | 0.0254 |
| O3 | 0.0029 | 0.0326 | 0.0250 | 0.0008 | - | - | 0.0000 |
| CO2 | - | - | - | - | 0.0003 | 0.0050 | 0.0142 |
| N2O | - | - | - | - | - | - | 0.0020 |
| CO | - | - | - | - | - | - | - |
| O2 | 0.0012 | 0.0010 | 0.0038 | 0.0000 | 0.0164 | - | - |
| NO | - | - | - | - | - | - | - |
| SO2 | - | - | - | - | - | - | - |
| NO2 | - | - | - | - | - | - | - |
| CH4 | - | - | - | - | 0.0000 | 0.0051 | 0.0003 |
| Total | 0.0042 | 0.0341 | 0.0344 | 0.0094 | 0.0216 | 0.0118 | 0.0420 |


Highlighted boxes show channels where total gas optical depth ≥0.02 to put in context the

requirement of aerosol optical depth accuracy of better than 0.02







**Table 2.2 : Optical depth of major atmospheric gases in 7 VIIRS channels**


| Channel  Gas | M3 | M4 | M5 | M7 | M8 | M10 | M11 |
|---|---|---|---|---|---|---|---|
| Wavelength (µm) | 0.488 | 0.551 | 0.67 | 0.861 | 1.239 | 1.601 | 2.257 |
| Gas | | | | | | | |
| H2O | 0.00009 | 0.00078 | 0.00066 | 0.00324 | 0.00844 | 0.00234 | 0.00542 |
| O3 | 0.00673 | 0.0312 | 0.01499 | 0.00075 | 0 | 0 | 0 |
| CO2 | 0 | 0 | 0 | 0 | 0.00041 | 0.02048 | 0.00001 |
| N2O | 0 | 0 | 0 | 0 | 0 | 0.00001 | 0.00403 |
| CO | 0 | 0 | 0 | 0 | 0 | 0 | 0 |
| O2 | 0.00184 | 0.00084 | 0.00144 | 0.00002 | 0.01147 | 0 | 0 |
| NO | 0 | 0 | 0 | 0 | 0 | 0 | 0 |
| SO2 | 0 | 0 | 0 | 0 | 0 | 0 | 0 |
| NO2 | 0 | 0 | 0 | 0 | 0 | 0 | 0 |
| CH4 | 0 | 0 | 0 | 0 | 0.00001 | 0.00085 | 0.04914 |
| Total | 0.00866 | 0.03282 | 0.01709 | 0.00401 | 0.02033 | 0.02368 | 0.0586 |


Highlighted boxes show channels where total gas optical depth ≥0.02 to put in context the

requirement of aerosol optical depth accuracy of better than 0.02




## Table 3.1: Gas Absorption Coefficients and Climatology for MODIS

| MODIS Band | Wavelength (µm) | Rayleigh Optical Depth | $O_3$ Optical Depth[#] | $H_2O$ Optical Depth[#] | Dry Gas[*] Optical Depth[#] | $O_3\_K0$ | $O_3\_K1$ | $H_2O\_K0$ | $H_2O\_K1$ | $H_2O\_K2$ |
|---|---|---|---|---|---|---|---|---|---|---|
| B3 | 0.4659 | 1.92E-01 | 2.90E-03 | 8.00E-05 | 1.25E-03 | -1.14E-04 | 8.69E-06 | -9.58E+00 | 1.23E+00 | -1.16E-01 |
| B4 | 0.5537 | 9.44E-02 | 3.26E-02 | 5.00E-04 | 9.50E-04 | 5.18E-06 | 9.50E-05 | -7.91E+00 | 1.00E+00 | -1.29E-02 |
| B1 | 0.6456 | 5.08E-02 | 2.52E-02 | 5.11E-03 | 3.91E-03 | 1.16E-04 | 7.32E-05 | -5.60E+00 | 9.40E-01 | -1.78E-02 |
| B2 | 0.8564 | 1.62E-02 | 8.10E-04 | 8.61E-03 | 2.00E-05 | 2.80E-07 | 2.36E-06 | -5.07E+00 | 8.77E-01 | -2.40E-02 |
| B5 | 1.2417 | 3.61E-03 | 0.00E+00 | 5.23E-03 | 1.69E-02 | 1.19E-07 | 1.55E-25 | -5.65E+00 | 9.81E-01 | -2.38E-02 |
| B6 | 1.6285 | 1.22E-03 | 0.00E+00 | 1.62E-03 | 9.98E-03 | 1.19E-07 | 5.17E-26 | -6.80E+00 | 1.03E+00 | -4.29E-03 |
| B7 | 2.1134 | 4.30E-04 | 2.00E-05 | 2.53E-02 | 1.63E-02 | 6.29E-07 | 7.03E-08 | -3.98E+00 | 8.86E-01 | -2.56E-02 |

\* Dry Gas includes $CO_2$, $CO$, $N_2O$, $NO_2$, $NO$, $CH_4$, $O_2$, $SO_2$
\# For each MODIS band, this nadir looking (viewing zenith angle = 0) optical depth
for the gas is computed from the US 1976 Standard Atmosphere in LBLRTM.



## Table 3.2: Gas Absorption Coefficients and Climatology for VIIRS

| VIIRS Band | Wavelength (μm) | Rayleigh Optical Depth | $O_3$ Optical Depth# | $H_2O$ Optical Depth# | Dry Gas* Optical Depth# | $O_3$_K0 | $O_3$_K1 | $H_2O$_K0 | $H_2O$_K1 | $H_2O$_K2 |
|---|---|---|---|---|---|---|---|---|---|---|
| M3 | 0.488 | 1.60E-01 | 6.73E-03 | 8.94E-05 | 1.84E-03 | -1.25E-04 | 1.98E-05 | -9.65E+00 | 9.87E-01 | 1.80E-04 |
| M4 | 0.5511 | 9.76E-02 | 3.11E-02 | 7.69E-04 | 8.34E-04 | -4.75E-05 | 9.08E-05 | -7.50E+00 | 9.84E-01 | -3.87E-03 |
| M5 | 0.6704 | 4.40E-02 | 1.50E-02 | 6.64E-04 | 1.44E-03 | -4.79E-05 | 4.37E-05 | -7.69E+00 | 9.95E-01 | -1.10E-02 |
| M7 | 0.8612 | 1.60E-02 | 7.70E-04 | 3.37E-03 | 2.45E-05 | 4.18E-07 | 2.24E-06 | -6.05E+00 | 9.65E-01 | -1.53E-02 |
| M8 | 1.2389 | 3.67E-03 | 0.00E+00 | 8.44E-03 | 1.19E-02 | 1.19E-07 | 5.17E-26 | -5.16E+00 | 9.59E-01 | -2.67E-02 |
| M10 | 1.6012 | 1.32E-03 | 0.00E+00 | 2.34E-03 | 2.13E-02 | 1.19E-07 | 1.03E-25 | -6.43E+00 | 1.02E+00 | -3.60E-03 |
| M11 | 2.257 | 3.50E-04 | 1.07E-06 | 5.42E-03 | 5.32E-02 | -2.61E-08 | 3.28E-09 | -5.85E+00 | 1.28E+00 | -5.04E-03 |

\* Dry Gas includes $CO_2$, $CO$, $N_2O$, $NO_2$, $NO$, $CH_4$, $O_2$, $SO_2$
\# For each VIIRS band, this nadir looking (viewing zenith angle = 0) optical depth for the gas is computed from the US 1976 Standard Atmosphere in LBLRTM.




## Table 4: Atmosphere Gas Correction Table Differences : C5 vs C6

| | C5 | C6 | Comment |
|---|---|---|---|
| RT Code | 6s | LBLRTM (Line-by-Line Radiative Transfer Model) | 6S is MODTRAN (Ref) database LBLRTM is HITRAN (Ref) database |
| # Gases Considered | 3 [$H_2O$, $O_3$, $CO_2$] | 10 [$H_2O$, $O_3$, $O_2$, $CO$, $CO_2$, $CH_4$, $NO$, $N_2O$, $NO_2$, $SO_2$ ] | Inclusion of 'other' dry gases in C6 created big differences in MODIS bands 5 & 7 (See Fig. 2) |
| Climatological GODs | Mid-latitude-Summer | US76 Standard Atmosphere | Ref |
| Rayleigh OD | Calculated for MODIS Bands's Fiter Function weighted Central λ (Sensor centroid) | Calculated for TOA centroid λ (solar irradiance and FF weighted) | Characteristic λ changes slightly |
| | ROD calculated based on single characteristic wavelength | ROD integrated over filter function | Corresponding slight change in sea level ROD |
