# Peer review of "Correcting for trace gas absorption when retrieving aerosol optical"

_Atmospheric Measurement Techniques, 2018_

## Referee Comment (RC1) · Anonymous Referee #1 · 5 Feb 2018

Important small detail for consistent climate data records

The paper discusses corrections to mid-visible satellite aerosol retrievals for the impact of the small but non-negligible trace gas absorption. It rightfully states that this well-known correction remains often at the side of publications on aerosol retrieval and not much detail is provided. The authors make a thorough quantitative assessment of the impact of different absorbing trace gases on window channels used in AOD retrieval. They relate the strength of absorption to typical AOD uncertainties and clearly show that while on global average trace gas absorption in window channels remains within AOD uncertainties, on regional scale it clearly does exceed them. The authors demon-

strate the importance of such an accurate trace gas correction for consistent climate data records by assessing the consequences of tiny differences between similar spectral channels of two similar but not identical sensors MODIS and VIIRS which can be used to constitute a long-term AOD record. This fully falls into the scope of AMT and sheds detailed light on such a "small" but still important aspect of long-term climate quality data records. The paper is well written with clear arguments and conclusions supported by substantial data presented in appropriate tables and figures. Title and abstract summarize clearly the content of the paper and its essence. The scientific methods used are state of the art and clearly referenced where suitable.

I therefore recommend to accept the paper after some technical corrections.

I have only three more general suggestions and a number of small issues (see detailed comments). 1) I recommend to consistently use the term "atmospheric gas correction" rather than "atmospheric correction" since the latter is commonly used when correcting surface observations for the impact of atmospheric scattering (molecules and aerosols) and trace gas absorption. 2) Regarding references I suggest to add a few references for leading European aerosol retrieval algorithms, mainly in context of the ESA Climate Change Initiative CCI: a) In addition to the reference to Hollmann et al. 2013 (overview paper of the entire CCI program with its 13 ECVs) a paper on the Aerosol project in CCI should be added: "Popp, et al., Development, Production and Evaluation of Aerosol Climate Data Records from European Satellite Observations (Aerosol_cci), Remote Sensing, 8, 421; doi:10.3390/rs8050421, 2016" or "de Leeuw, et al., Evaluation of seven European aerosol optical depth retrieval algorithms for climate analysis, Remote Sensing of Environment, 162, 295-315, doi: 10.1016/j.rse.04.023, 2015" b) When referring to the GCOS requirements, the latest GCOS implementation plan 2016 with its Annex A ECV Product requirement tables (aerosol on page 281) should be added: GCOS-IP 2016: GCOS Implementation Plan 2016. GCOS-200. Available at https://library.wmo.int/opac/doc_num.php?explnum_id=3417 c) As leading algorithm for the AATSR instrument the Swansea algorithm should be added to the references:

"Bevan, et al. (2012), A global dataset of atmospheric aerosol optical depth and surface reflectance from AATSR. Remote Sensing of Environment, 116, 119-210" and "North, et al., Retrieval of land surface bidirectional reflectance and aerosol opacity from ATSR-2 multiangle imagery, IEEE Trans. Geosci. Remote Sens. 1999, 37, 526–537. 3) Regarding figures, I suggest to increase axis legends of fig. 1, colour bar legends of fig. 6a

Detailed comments: l. 82: "[MISR ATBD]" – please add http link, date and version to make a unique reference l. 111: please also add the width of VIIRS channels l. 178 / 179: two conflicting namings for Gi are made (airmass factor or path length), which could be confusing readers l. 270 ff: here ten gases are discussed, while in l. 169 only 5 are identified as relevant – please harmonize or explain (see also l. 346 and 462/463) l. 304 ff: I had to read this several times and found it confusing (my first impression was that liner relationship will do then is learned that it will not do) – maybe you can rearrange to start with the clear statement that a quadratic fit is needed l. 310 ff: Can you state whether the H2O airmass factors include any effect of multiple scattering (since water vapour prevails in the lower troposphere)? l. 318: instead of "Gi" and "gas i" it should be "GO3" and "O3" l. 332: What is the temporal resolution of the NCEP analysis? l. 386: table 4 contains also another line on Rayleigh OD, which should also be mentioned and brought into perspective in the text l. 415: correct in the middle: "that that" l. 486: Another aspect of interest would be whether an overlooked long-term trend in water vapour or ozone concentrations (e.g. by using a static climatology) could create an artificial AOD trend – if you could add a statement on this (whether its relevant or negligible) based on an assumed possible decadal trend in concentrations, this would be very useful? Fig. 7 legend vs. text in l. 427: differences are smaller than 0.08 (figure) or 0.07 (text) – can you harmonize? Table 4 / Rayleigh OD, 2nd column: correct "Fiter" to "Filter"

---

## Short Comment (SC1) · 14 Feb 2018

Why HITRAN2008 is used? There are substantial improvements in the quality and extent of the spectroscopic data for atmospheric gases in HITRAN2016 (Gordon et al, J. Quant. Spectrosc. Radiat. Transf. (2017) 203, 3-69. doi:10.1016/j.jqsrt.2017.06.038) and even HITRAN2012 (Rothman et al, J. Quant. Spectrosc. Radiat. Transf. 130, 4-50 (2013) doi:10.1016/j.jqsrt.2013.07.002). In fact many trace gases did not even have data in the shortwave regions in HITRAN2008. In addition substantial improvements in the quality of the spectroscopic data and its completeness for main absorbers, including water vapor and ozone were carried out in more recent editions. HITRAN2016

data is available at www.hitran.org

---

## Referee Comment (RC2) · A.C. Povey (Referee) · 20 Feb 2018

This paper outlines a revision of the Dark Target algorithm to better account for the absorption of common atmospheric gases. The radiative transfer code and spectral database have been updated since Collection 5, but the essence of the method is unchanged. A gas's optical path is modelled as a linear (or, in the case of water, quadratic) function of gas path length. Ten gases were evaluated, and the regression coefficients are reported for both the MODIS and VIIRS sensors. It concludes with a brief reminder that ignoring the difference in spectral response between two instruments can produce non-trivial errors.

Though this paper has a rather small audience, it is well within the remit of this journal and is the sort of work that is often overlooked. Other than a few minor corrections, I recommend it for publication.

A few matters that warrant consideration:

L28 Though I appreciate the simplicity of the language, aerosols aren't necessarily 'tiny'. Hinds described them as 'fine' but I find 'Aerosols are particles in the atmosphere' is usually sufficient.

L285 It is true that these gases are usually well mixed. However, a not-insignificant number of users of aerosol data study emissions from volcanoes and fires. As those emit many of the gases you are studying in significant quantities, do you have any estimate of the magnitude of errors that will result from using climatological concentrations there? A back-of-the-envelope calculation could be quite informative.

L304-9 Could you please provide some quantitative indication of the quality and uncertainty of these fits (e.g. root-mean-square deviation and the maximum error)? This would be particularly instructive for water vapour, where I would appreciate a more scientific justification for using a quadratic fit (or, *vise versa*, a justification for using linear fits with everything else).

L336-8 While I appreciate that in normal operations you can't use the MODIS water vapour product, you've presumably tried using it offline. Could you quantify approximately how much difference it makes to the final product?

L441-3 This final paragraph begs the obvious question to any algorithm paper: You've proposed something that sounds sensible, but is it actually better than what you did before? Fig. 7 implies you've processed at least a month of data with the new corrections. For that data, does the RMS difference against AERONET collocations improve (or at least not significantly degrade)?

Fig.1 The axes labels are far too small to be legible.

Figs.3-4 The axes labels aren't meaningful to someone that hasn't read the text exhaustively. Also, is the $y$-axis of 3(a) really the log of the log of the transmission factor?

Fig.6(a) Could this be the same size as 6(b) to facilitate comparison?

Fig.7 There might be a good reason why not, but could the fractional difference be plotted rather than (or in addition to) the absolute difference? Over the central Pacific changes appear to be 0.01, which is rather significant there.

The English quality is in the upper quartile of paper's I've reviewed. Though I found the language rather repetitive, it is not my place to nitpick style. However, I do have some grammatical recommendations:

- "the" should proceed the following words: L12 underlying, L14 Moderate, L18 High-resolution, L21 MODIS, L25 gas, L34 characterization, L40 solar, L53 accuracy, L98 context, L106 Earth's, L126 spectral reflectance, L211 coefficients, L311 gas, L326 largest, L381 HITRAN, L395 subsequent, L400 MODIS, L470 nadir.

L17 There should be a comma after 'paper'.

L23 AOD biases of up to

L24 studies are attempting have attempted to create

L35 'over broad regions' seems redundant to 'global'.

L41 from the solar radiation interacting interaction with suspended aerosol particles aerosols from the

L51 to apply in to new situations. These latter include

L55 suggested that  to reduce uncertainties

L67 magnitude  to pristine AOD, and is equal

L76 (well-mixed)  across the globe

L77 their absorption would also  lead to

L82 gases to be  negligible [MISR ATBD]

L105 from blue  to the shortwave

L116 'observed' seems redundant to 'as measured by the satellite'.

L123 been made  about the surface

L141 each gas  was calculated

L195 The expression for $G$ should be typeset as math not text.

L252 This heading should be bold.

L256 database []  to calculate transmittance

L267 This heading should be bold.

L323 accordance  with absorption

L329 $w$ should be typeset as math not text.

L347 $SO_2$, and other trace

L348 day-to-day should be hyphenated.

L355 different gases is different
L357 case of with a small amount

L371 'match' seems to be the wrong word. I think you mean 'be consistent with' or 'can be used with'.

Fig.1 are overlaid for visualizing to visualize their positioning in atmospheric 'window' region regions where

There are also a few thoughts I would like the authors to be aware of but which are unreasonable to expect a revision:

L86 Though aerosol retrievals don't often discuss gas correction, sea surface temperature studies do because of their more stringent accuracy requirements, such as doi:10.1016/j.rse.2010.10.016. For aerosol in particular, §2.3.3.3 of the thesis of Haiyan Huang (http://eodg.atm.ox.ac.uk/eodg/theses/Huang.pdf and https://ora.ox.ac.uk/objects/uuid:16e444e6-5da9-43da-a122-c50c7e6a2412) presents a sensitivity study of TOA brightness temperature from AATSR to a variety of gases. I am curious if the authors have ever considered the importance of species such as $F^{12}$ and CFCs, which Dr. Huang found to be rather important?

L257 I agree with Dr. Gordon that you should have used a more recent version of HITRAN.

L281 Many studies desire a representative set of atmospheric profiles. I appreciate that you've cited someone for making that decision. However, researchers have done statistically robust selections for the minimally representative set. For example, §3 of https://www.ecmwf.int/sites/default/files/elibrary/2008/11040-generation-rttov-regression-coefficients-iasi-and-airs-using-new-profile-training-set-and-new.pdf.

---

## Author Response (AR1)

Important small detail for consistent climate data records

The paper discusses corrections to mid-visible satellite aerosol retrievals for the impact of the small but non-negligible trace gas absorption. It rightfully states that this wellknown correction remains often at the side of publications on aerosol retrieval and not much detail is provided. The authors make a thorough quantitative assessment of the impact of different absorbing trace gases on window channels used in AOD retrieval. They relate the strength of absorption to typical AOD uncertainties and clearly show that while on global average trace gas absorption in window channels remains within AOD uncertainties, on regional scale it clearly does exceed them. The authors demonstrate the importance of such an accurate trace gas correction for consistent climate data records by assessing the consequences of tiny differences between similar spectral channels of two similar but not identical sensors MODIS and VIIRS which can be used to constitute a long-term AOD record. This fully falls into the scope of AMT and sheds detailed light on such a "small" but still important aspect of long-term climate quality data records. The paper is well written with clear arguments and conclusions supported by substantial data presented in appropriate tables and figures. Title and abstract summarize clearly the content of the paper and its essence. The scientific methods used are state of the art and clearly referenced where suitable.

I therefore recommend to accept the paper after some technical corrections.

Thank you very much for a thorough review and for suggestions that have improved the paper. We have incorporated all of them.

I have only three more general suggestions and a number of small issues (see detailed comments).
1) I recommend to consistently use the term "atmospheric gas correction" rather than "atmospheric correction" since the latter is commonly used when correcting surface observations for the impact of atmospheric scattering (molecules and aerosols) and trace gas absorption.

Done

2) Regarding references I suggest to add a few references for leading European aerosol retrieval algorithms, mainly in context of the ESA Climate Change Initiative CCI: a) In addition to the reference to Hollmann et al. 2013 (overview paper of the entire CCI program with its 13 ECVs) a paper on the Aerosol project in CCI should be added: "Popp, et al., Development, Production and Evaluation of Aerosol Climate Data Records from European Satellite Observations (Aerosol_cci), Remote Sensing, 8, 421; doi:10.3390/rs8050421, 2016" or "de Leeuw, et al., Evaluation of seven European aerosol optical depth retrieval algorithms for climate analysis, Remote Sensing of Environment, 162, 295-315, doi: 10.1016/j.rse.04.023, 2015"
Added b) When referring to the GCOS requirements, the latest GCOS implementation plan 2016 with its Annex A ECV Product requirement tables (aerosol on page 281) should be added: GCOS-IP 2016: GCOS Implementation Plan 2016. GCOS-200. Available at https://library.wmo.int/opac/doc_num.php?explnum_id=3417

Added c) As leading algorithm for the AATSR instrument the Swansea algorithm should be added to the references:
"Bevan, et al. (2012), A global dataset of atmospheric aerosol optical depth and surface reflectance from AATSR. Remote Sensing of Environment, 116, 119-210" and
"North, et al., Retrieval of land surface bidirectional reflectance and aerosol opacity from ATSR-2 multiangle imagery, IEEE Trans. Geosci. Remote Sens. 1999, 37, 526–537.
Added

3) Regarding figures, I suggest to increase axis legends of fig. 1, colour bar legends of fig. 6a
Done

**Detailed comments:**
l. 82: "[MISR ATBD]" – please add http link, date and version to make a unique reference
Done l. 111: please also add the width of VIIRS channels
Done l. 178/ 179: two conflicting namings for Gi are made (airmass factor or path length), which could be confusing readers
Corrected l. 270 ff: here ten gases are discussed, while in l. 169 only 5 are identified as relevant – please harmonize or explain (see also l. 346 and 462/463)
Done l. 304 ff: I had to read this several times and found it confusing (my first impression was that liner relationship will do then is learned that it will not do) – maybe you can rearrange to start with the clear statement that a quadratic fit is needed
Thanks for pointing this out. Rearranged.

l. 310 ff: Can you state whether the $H_2O$ airmass factors include any effect of multiple scattering (since water vapour prevails in the lower troposphere)?
LBLRTM does not model multiple scattering. The only scattering effect modeled in LBLRTM is Rayleigh extinction.

l. 318: instead of "Gi" and "gas i" it should be "GO3" and "O3"
Corrected. Thanks.

l. 332: What is the temporal resolution of the NCEP analysis?
It is 6 hrs. Information in Line 374

l. 386: table 4 contains also another line on Rayleigh OD, which should also
be mentioned and brought into perspective in the text
It has been edited out since this information is not relevant to this paper and was inadvertently left
in the table l. 415: correct in the middle: "that that"
Done l. 486: Another aspect of interest would be whether an overlooked longterm
trend in water vapour or ozone concentrations (e.g. by using a static climatology)
could create an artificial AOD trend – if you could add a statement on this (whether its
relevant or negligible) based on an assumed possible decadal trend in concentrations,
this would be very useful?

For $H_2O$ and $O_3$, we almost never need to use climatology since we use NCEP data and that can
account for trend from these gases. However, for the 'dry gas' we use climatology. For the DT
algorithm, the important gases in category are $CO_2$ (MODIS) and $CH_4$ (VIIRS).

Over the lifetime of MODIS instrument, $CO_2$ has increased from about 375 to 405 (i.e by ~10%).
A 10% change in $CO_2$ amount = $10* 0.0163 = 0.00163$ $CO_2$ Optical depth at 2.12 μm (MODIS
Channel impacted by $CO_2$ absorption). Not accounting for this trend in atmospheric gas
correction means that the DT algorithm would retrieve lower AODs and smaller trends. In
separate work, we have also estimated the uncertainty in the retrieved AOD due to error in the
water vapor data used by the DT algorithm. A 20% error in water vapor content results in AOD
uncertainty of ~0.002 (median) , ~0.003 (mean). The uncertainty magnitude was similar for
different months of global data we looked at. Therefore, we think that the impact of using
climatological values for 'dry gases' would have an overall negligible impact on the decadal
trends.

Fig. 7 legend vs. text in l. 427: differences are smaller than 0.08 (figure) or 0.07 (text) – can you
harmonize?
Corrected.

Table 4 / Rayleigh OD, 2nd column: correct "Fiter" to "Filter"
Done

Atmos. Meas. Tech. Discuss.,
doi:10.5194/amt-2018-7-RC2, 2018
This paper outlines a revision of the Dark Target algorithm to better account for the absorption of common atmospheric gases. The radiative transfer code and spectral database have been updated since Collection 5, but the essence of the method is unchanged. A gas's optical path is modelled as a linear (or, in the case of water, quadratic) function of gas path length. Ten gases were evaluated, and the regression coefficients are reported for both the MODIS and VIIRS sensors. It concludes with a brief reminder that ignoring the difference in spectral response between two instruments can produce non-trivial errors.

Though this paper has a rather small audience, it is well within the remit of this journal and is the sort of work that is often overlooked. Other than a few minor corrections, I recommend it for publication.

Thank you very much for a thorough review and for suggestions that have improved the paper. We have incorporated all of them.

A few matters that warrant consideration:
L28 Though I appreciate the simplicity of the language, aerosols aren't necessarily 'tiny'. Hinds described them as 'fine' but I find 'Aerosols are particles in the atmosphere' is usually sufficient.

We have replace 'tiny' with 'fine'.

L285 It is true that these gases are usually well mixed. However, a not-insignificant number of users of aerosol data study emissions from volcanoes and fires. As those emit many of the gases you are studying in significant quantities, do you have any estimate of the magnitude of errors that will result from using climatological concentrations there? A back-of-the-envelope calculation could be quite informative.

We agree that the climatological concentration of gases is not representative of their concentrations at sources such as volcanoes, fires, industries etc. We are using climatological optical depth for $CO_2$, $N_2O$, $O_2$ and $CH_4$. Some of these gases are emitted by fires ($CO_2$, $N_2O$, and $CH_4$) and volcanoes ($CO_2$, $SO_2$). These gases are transparent to solar radiation in the shortwave channels used for aerosol retrieval. Tables 2.1, 2.2 show that these gases absorb in the 1.6 μm and the 2.1 μm channels.

CO2 concentrations (AIRS Maps) indicate ~2% [400 ppm to 408 ppm] increase in fire locations compared to background. In the 2.1 μm channel, used by DT algorithm, 2% increase in CO2 concentration would translate to ~0.001 CO2 optical depth, which would not change the total gas optical depth in 2.1 μm channel by a lot.

Similarly, from CH4 maps from volcano events, there seems to range by ±65 ppbv (~3.5%). For VIIRS, CH4 optical depth in 2.1 μm channel (0.04914) will change by ~0.002.

So, using climatology for gases other than H2O and O3, will impact the atmospheric gas corrections near source regions. However, since these gases absorb in IR, where signal is low, the errors will be small.

L304-9 Could you please provide some quantitative indication of the quality and uncertainty of these fits (e.g. root-mean-square deviation and the maximum error)?
To-Do

This would be particularly instructive for water vapour, where I would appreciate a more scientific justification for using a quadratic fit (or, vise versa, a justification for using linear fits with everything else).

Thank you for this comment. We have revised the text with additional information :
The quadratic nature seems to be rooted in the absorption characteristics of water vapor. Water vapor absorbs over a wide range of electromagnetic spectrum. In different regions of the spectrum the molecule absorbs radiation via different absorption mechanism (rotational transitions and intermolecular vibrations in microwave and IR, intramolecular vibrational transitions in IR and electronic transitions in UV). The spectra are also all different for isotopic variants of water (e.g. HDO, $D_2O$, $H_2{}^{18}O$) and is also a function of pressure and temperature. It is also found that with greater levels of water vapor the increase in water dimer will be nearly quadratic, and increase the direct absorption of sunlight to a greater extent, than water monomer absorption and line broadening.  The vertical distribution of water vapor in the atmosphere may result in the quadratic nature of absorption/transmission relationship depicted by RT calcuations.

L336-8 While I appreciate that in normal operations you can't use the MODIS water vapour product, you've presumably tried using it offline. Could you quantify approximately how much difference it makes to the final product?

We haven't used the MODIS water vapor product. However, we have estimated the uncertainty in the retrieved AOD due to error in the water vapor data used by the DT algorithm. A 20% error in water vapor content results in AOD uncertainty of ~0.002 (median) , ~0.003 (mean). The uncertainty magnitude was similar for different months of global data we looked at.

L441-3 This final paragraph begs the obvious question to any algorithm paper: You've proposed something that sounds sensible, but is it actually better than what you did before? Fig. 7 implies you've processed at least a month of data with the new corrections. For that data, does the RMS difference against AERONET collocations improve (or at least not significantly degrade)?

We had tested to ensure that our empirical relations (Eqs 5, 6) can reproduce the transmission calculated from LBLRTM for different water vapor and ozone profiles. At the time of doing this work, few other changes were made in the retrieval algorithm. The

Fig.1 The axes labels are far too small to be legible.
Corrected

Figs.3-4 The axes labels aren't meaningful to someone that hasn't read the text exhaustively. Also, is the y-axis of 3(a) really the log of the log of the transmission factor?

Thank you. Added more description. Yes, y-axis label is correct

Fig.6(a) Could this be the same size as 6(b) to facilitate comparison?
Done

Fig.7 There might be a good reason why not, but could the fractional difference be plotted rather than (or in addition to) the absolute difference? Over the central Pacific changes appear to be 0.01, which is rather significant there.
Done. Additional figure added

The English quality is in the upper quartile of paper's I've reviewed. Though I found the language rather repetitive, it is not my place to nitpick style. However, I do have some grammatical recommendations:

All edits suggested here have been incorporated in the paper. Thank you for the suggestions.

• "the" should proceed the following words: L12 underlying, L14 Moderate, L18 High-resolution, L21 MODIS, L25 gas, L34 characterization, L40 solar, L53 accuracy, L98 context, L106 Earth's, L126 spectral reflectance, L211 coefficients, L311 gas, L326 largest, L381 HITRAN, L395 subsequent, L400 MODIS, L470 nadir.
L17 There should be a comma after 'paper'.
L23 AOD biases of up to
L24 studies are attempting have attempted to create
L35 'over broad regions' seems redundant to 'global'.
L41 from the solar radiation interacting interaction with suspended aerosol particles aerosols from the

L51 to apply in to new situations. These latter include

L55 suggested that for reducing to reduce uncertainties

L67 magnitude as to pristine AOD, and is equal

L76 (well-mixed) throughout across the globe

L77 their absorption would also would lead to

L82 gases to be too small to bother with negligible [MISR ATBD]

L105 from blue through to the shortwave

L116 'observed' seems redundant to 'as measured by the satellite'.

L123 been made for about the surface

L141 each gas were was calculated

L195 The expression for G should be typeset as math not text.

L252 This heading should be bold.

L256 database [] for calculating to calculate transmittance

L267 This heading should be bold.

L323 accordance to with absorption

L329 w should be typeset as math not text.

L347 $SO_2$, and other trace

L348 day-to-day should be hyphenated.

L355 different gases is different

L357 case of with a small amount

L371 'match' seems to be the wrong word. I think you mean 'be consistent with' or 'can be used with'.

Fig.1 are overlaid for visualizing to visualize their positioning in atmospheric 'window' region regions where

All above edits have been incorporated in the paper. Thank you for the suggestions.

**There are also a few thoughts I would like the authors to be aware of but which are unreasonable to expect a revision:**

L86 Though aerosol retrievals don't often discuss gas correction, sea surface temperature studies do because of their more stringent accuracy requirements, such as doi:10.1016/j.rse.2010.10.016. For aerosol in particular, §2.3.3.3 of the thesis of Haiyan Huang (http://eodg.atm.ox.ac.uk/eodg/theses/Huang.pdf and https://ora.ox.ac.uk/objects/uuid:16e444e6-5da9-43da-a122-c50c7e6a2412) presents a sensitivity study of TOA brightness temperature from AATSR to a variety of gases. I am curious if the authors have ever considered the importance of species such as $F_{12}$ and CFCs, which Dr. Huang found to be rather important?

Dr Huang found these gases to be important in the Infrared channels. The DT algorithm doesn't use these channels for aerosol retrievals. For the channels used in DT algorithm, we have looked at all those gases (in HITRAN database) that might have some absorption in these channels. Only the ones listed in the paper are the important ones.

L257 I agree with Dr. Gordon that you should have used a more recent version of HITRAN.

We agree. The only reason for using the earlier version of HITRAN was because at the time when modifications were made to our gas tables (2012 – 2013), HITRAN2008 was the latest version. In the next version of DT algorithm, we will update our calculations with latest HITRAN version.

L281 Many studies desire a representative set of atmospheric profiles. I appreciate
that you've cited someone for making that decision. However, researchers
have done statistically robust selections for the minimally representative
set. For example, §3 of https://www.ecmwf.int/sites/default/files/elibrary/2008/
11040-generation-rttov-regression-coefficients-iasi-and-airs-using-new-profile-training-set-and-
new.
pdf.

Thank you for pointing to this reference. We have added the relevant reference to the paper.

Why HITRAN2008 is used? There are substantial improvements in the quality and extent of the spectroscopic data for atmospheric gases in HITRAN2016 (Gordon et al, J. Quant. Spectrosc. Radiat. Transf. (2017) 203, 3-69. doi:10.1016/j.jqsrt.2017.06.038) and even HITRAN2012 (Rothman et al, J. Quant. Spectrosc. Radiat. Transf. 130, 4-50 (2013) doi:10.1016/j.jqsrt.2013.07.002). In fact many trace gases did not even have data in the shortwave regions in HITRAN2008. In addition substantial improvements in the quality of the spectroscopic data and its completeness for main absorbers, including water vapor and ozone were carried out in more recent editions. HITRAN2016 data is available at www.hitran.org

Thank you for your comment. We agree with you. The only reason for using HITRAN2008 in this study is because at the time when modifications were made to our gas tables (2012 – 2013), HITRAN2008 was the latest version. In the next version of DT algorithm, we will update our calculations with the latest HITRAN version.

[revised manuscript text omitted]